# Bio-inspired reversible underwater adhesive

Yanhua Zhao[1], Yang Wu[1,2], Liang Wang[1,3], Manman Zhang[1], Xuan Chen[1], Minjie Liu[1], Jun Fan[4], Junqiu Liu [3], Feng Zhou[2] & Zuankai Wang [1,5]

The design of smart surfaces with switchable adhesive properties in a wet environment has remained a challenge in adhesion science and materials engineering. Despite intense demands in various industrial applications and exciting progress in mimicking the remarkable wet adhesion through the delicate control of catechol chemistry, polyelectrolyte complex, and supramolecular architectures, the full recapitulation of nature's dynamic function is limited. Here, we show a facile approach to synthesize bioinspired adhesive, which entails the reversible, tunable, and fast regulation of the wet adhesion on diverse surfaces. The smart wet adhesive takes advantage of the host–guest molecular interaction and the adhesive nature of catechol chemistry, as well as the responsive polymer, allowing for screening and activation of the interfacial interaction simply by a local temperature trigger in an on-demand manner. Our work opens up an avenue for the rational design of bioinspired adhesives with performances even beyond nature.

[1] Department of Mechanical and Biomedical Engineering, City University of Hong Kong, Hong Kong 999077, Hong Kong. [2] Lanzhou Institute of Chemical Physics, Chinese Academy of Sciences, Lanzhou 73000, China. [3] State Key Laboratory of Supramolecular Structure and Materials, College of Chemistry, Jilin University, 2699 Qianjin Road, Changchun 130012, China. [4] Department of Physics and Materials Science, City University of Hong Kong, Hong Kong 999077, Hong Kong. [5] Shenzhen Research Institute of City University of Hong Kong, Shenzhen 518057, China. Yanhua Zhao, Yang Wu, and Liang Wang contributed equally to this work. Correspondence and requests for materials should be addressed to F.Z. (email: zhouf@licp.cas.cn) or to Z.W. (email: zuanwang@cityu.edu.hk)

Our daily life experience has that the strong interaction manifested in commercial glues, and target surfaces in a dry environment, are prone to be disrupted in the presence of water[1, 2]. The breakdown of the contact adhesion as a result of adsorption of water molecules between their interfaces impacts a wide range of areas, such as water pipeline leakage, corrosion of hull, and artificial teeth take off. In addition to sustaining a strong adhesion under the aqueous environment, in many practical applications, adhesive materials that are capable of adhering to target surfaces in on-demand manner are highly preferred to prevent indiscriminate sticking. Thus, one of the great challenges in the adhesion science and technology is to develop smart materials with switchable adhesive properties in a wet environment[3].

In nature, many marine organisms such as mussels, sandcastle worms, and barnacles orchestrate a powerful wet adhesion mechanism through the elegant blend of catechol chemistry, polyelectrolyte complex, and supramolecular architectures[1–9]. Over the past decade, it has been shown that the remarkable wet adhesion tenacity can be mimicked or emulated by the use of 3,4-dihydroxy-L-phenylalanine (DOPA) groups[10–12]. Since this seminal work, extensive progress has been made in the understanding of basic mechanisms underlying the intriguing interaction, as well as in the design of artificial adhesives for multifunctional applications[9, 11, 13–21]. Despite exciting progress, the capacity to recapitulate nature's functional dynamics remains limited[15]. Unlike in the dry condition where the reversible interfacial adhesion can be achieved through the external stimulus such as pH, light, temperature, humidity, and magnetic force or through the modulation of the bulk material properties[22–27], the attainment of the reversible adhesion in the wet environment remains extremely elusive due to the complex interfacial interaction[28–30]. Thus, a robust and smart underwater adhesive which endows a tunable, reversible modulation and can be deposited on a wide range of substrates is yet to be developed.

Herein, we report on a viable approach to fabricating a bioinspired adhesive that renders a reversible, dynamic, and fast regulation of underwater adhesion. Our strategy leverages on the synergistic cooperation of catechol chemistry, responsive wettability, as well as selective host–guest interaction, which confers control over properties in a reversible, highly tunable, and dynamic fashion. Moreover, we demonstrate that our biomimetic adhesive can be applied to various substrates and displays superior adhesion properties.

## Results

**Design and synthesis of the wet adhesive.** Our adhesive mainly consists of two copolymers: a mussel-inspired guest-adhesive copolymer and a thermoresponsive host copolymer. We start with conjugating the mussel-inspired adhesive DOPA polymer with the guest motif adamantine (AD) and methoxyethyl acrylate (MEA) monomer to form a guest copolymer, referred to as pDOPA-AD-MEA, in which DOPA serves as the adhesive moiety, AD as the guest moiety that can specially combine with host moiety via host–guest recognition, and MEA as the hydrophobic matrix to enhance the adhesion property of DOPA. To reversibly regulate the wet adhesion property, we further create the host copolymer pNIPAM-CD, in which the poly(N-isopropylacrylamide) (pNIPAM) is a smart and responsive polymer well known for its reversible conformational transition as the temperature changes around its lower critical solution temperature (LCST) and β-cyclodextrin (β-CD) is the host molecule designed for the selective binding with the guest AD moiety in the guest pDOPA-AD-MEA copolymer[31, 32]. Finally, we use the dip-coating process to deposit the as-prepared guest copolymer on a silicon substrate, followed by the self-assembly of the host copolymer through the host–guest chemistry (Fig. 1a). The detailed fabrication procedure and characterization of the host and guest copolymers are shown in Methods section and Supplementary Figs. 1–3. The LCST of as-synthesized adhesive is dictated by the pNIPAM to CD monomer proportion in the host copolymer (pNIPAM-CD). For our adhesive coating, the LCST is measured to be ~35 °C.

The successful decoration of the adhesive coating on silicon substrate is evidenced by chemical composition analysis using X-ray photoelectron spectroscopy (XPS) and attenuated total reflection infrared spectroscopy (AT-IR, Methods section). As shown in the XPS spectra (Fig. 1b), intense peaks at 99.8 and 150.1 eV are ascribed to Si 2p and 2s signals for the bare silicon substrate. After coating the adhesive guest copolymer pDOPA-AD-MEA on silicon, the Si signals vanish and instead a weak N 1s peak at 399.5 eV emerges, which is assigned to the DOPA group. By contrast, the N 1s peak observed on the adhesive coating becomes much stronger after pNIPAM-CD attachment, indicating the successful assembly of pNIPAM-CD onto the adhesive guest copolymer through the host–guest interaction (Fig. 1c). Careful inspection of the content ratios of N element in the wet adhesive relative to that in the guest copolymer also confirms the successful self-assembly of stimuli-responsive host copolymer (Supplementary Fig. 4 and Supplementary Table 1). In addition, the ATIR spectra measurement clearly shows the presence of characteristic peaks of DOPA and pNIPAM (Fig. 1d), which further demonstrates the successful synthesis of the adhesive coating.

**Adhesion measurement.** Next, we measure the wet adhesion property of the as-prepared adhesive coating using an atomic force microscope (AFM) decorated with a temperature controller (Methods section). Under a local temperature above LCST (40 °C in this case), the adhesion force is measured to be ~23 nN (Fig. 2a). However, when the adhesive surface temperature is decreased below LCST (25 °C in this case), the interfacial adhesion is deactivated and gives a minimal force of ~2.2 nN (Fig. 2b). The distinct change in the adhesion force in response to different temperatures is also reflected by the large difference in their retraction distance. Notably, upon raising the local temperature of the adhesive above LCST again, the interfacial adhesion is reactivated, exhibiting a full reversible signature. In addition, there is no notable decay in the adhesion strength based on continuous measurements spanning 10 cycles (Fig. 2c).

The remarkable microscopic adhesion property can be translated into exceptional adhesion at the large scale as well. For macroscopic characterization, we deposit a uniform layer of adhesive ~90 μm in thickness onto a silicon substrate with a size of 1.5×1.5 cm, and measure the underwater adhesion using a highly sensitive universal testing machine (Supplementary Fig. 5). The macroscopic adhesion strength of the adhesive coating against the target surface at 40 °C is ~20 folds larger than that at 25 °C (Fig. 2d, e), which is consistent with that obtained in the AFM characterization. Using the method reported in previous studies[33, 34], the work of adhesion of our wet adhesive coating at 40 °C is calculated to be 4370 mJ m$^{-2}$, which is much larger than that at 25 °C (182 mJ m$^{-2}$). The adhesion strength is also five times larger than that of the 3 M double-sided tape underwater (Supplementary Fig. 6). Notably, the wet adhesion can be further amplified by increasing preloading or contact time to repel the surrounding water film between the adhesive and the adhered surface (Supplementary Fig. 7). In addition to superior macroscopic adhesion capability, in consistence with the microscopic measurement, there is no marked decay observed in the adhesion

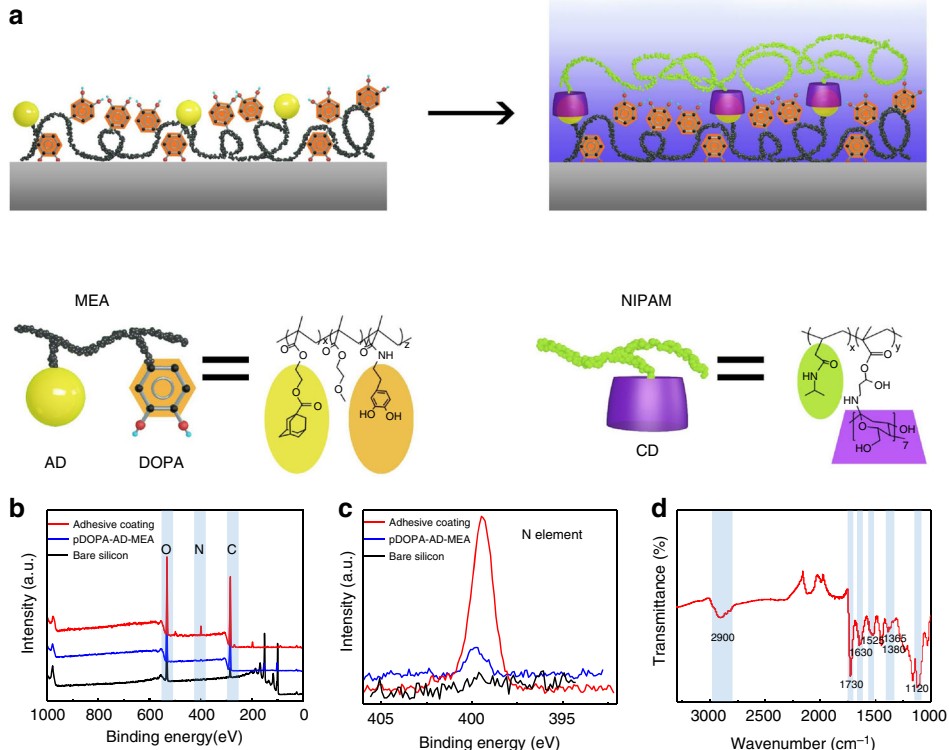

**Fig. 1** Bioinspired wet adhesive synthesis and characterization. **a** The schematic diagram showing the specific procedure for the preparation of the wet adhesive. The synthesis process involves the creation of an adhesive guest copolymer pDOPA-AD-MEA on a clean Si substrate using the dip-coating method, followed by the self-assembly of host copolymer pNIPAM-CD using the host–guest molecular recognition. **b, c** X-ray photoelectron spectroscopy (XPS) analysis of the chemical composition of the silicon surface, adhesive guest copolymer pDOPA-AD-MEA, as well as the adhesive coating, demonstrating the successful synthesis of the as-prepared adhesive. **d** The AT-IR spectra measurement of the adhesive coating. The characteristic peak at 1525 cm$^{-1}$, is assigned to the stretching vibration of C=C of the aromatic ring on DOPA; the peak at 1630 cm$^{-1}$ is assigned to the stretching vibration of C=O bond on NIPAM; and the peaks at 1380 cm$^{-1}$ and 1365 cm$^{-1}$ are assigned to double-methyl groups on pNIPAM. The above results demonstrated the presence of DOPA and pNIPAM

strength after several cycles of measurement, confirming that the wet adhesion is indeed fully reversible (Fig. 2f). In particular, the adhesive coating can maintain a stable adhesion for more than 36 h in the wet environment, indicating its superior stability (Supplementary Fig. 8). Although the specific response time to switch the interfacial adhesion is not quantitatively measured, we believe that it is only limited by the speed in temperature control. Taken together, compared to previous approaches such as chemical control, the adhesion in our adhesive can be dynamically mediated in a more flexible and faster manner[3, 14].

In practical applications, the underwater adhesive might be operated under different wet conditions. Therefore, we further investigate how the adhesion strength is influenced by pH and ion types of solutions. The adhesion strength of the adhesive in solutions with pH values ranging from 2.26 to 10.66 is given in Fig. 3a. It is clear that the adhesion strength at low pH values such as 2.26, 3.21, 4.32, and 5.22 is comparable. When the pH value increases to 9.51, ~50% reduction in the adhesion strength is measured. Moreover, for the coating immersed in the solution with pH of 10.66, there is no pronounced adhesion between two surfaces. The failure of wet adhesion in alkaline solutions might be due to the oxidation of the adhesive catechol[35, 36]. We also compare the adhesion strength of the wet adhesive coating in different ionic solutions including NaCl, KCl, CaCl$_2$, MgCl$_2$, ZnCl$_2$, and FeCl$_3$, all of which have a concentration of 1 M. As shown in Fig. 3b, there is no notable adhesion at 40 °C for the adhesive immersed in FeCl$_3$ solution. By contrast, for other cases, adhesion strengths are approximately identical (~4 kPa), which

are also dramatically larger than that in the case of FeCl$_3$. The loss of wet adhesion at the solution containing Fe$^{3+}$ ion might be ascribed to the chemical binding of Fe$^{3+}$ with DOPA moiety and subsequent formation of Fe(DOPA)$_3$ cross-links[37–39].

To confirm whether the intriguing wet adhesion behavior is indeed resulting from the adhesive moiety (DOPA), we further compare the adhesion property of our adhesive with a control sample. The control surface is constructed by the deposition of a guest copolymer AD-MEA on silicon substrate without the introduction of DOPA (AD-MEA), followed by the assembly of host copolymer pNIPAM-CD via the host–guest recognition. As illustrated in Supplementary Fig. 9a, the adhesion strength of the control sample is ~1.8 kPa, which is much smaller than that of its counterpart. Moreover, a significant decay in the adhesion strength can be observed after successive testing cycles. After ~50 cycles of measurement, the adhesion strength almost decreases to zero. However, AD-MEA-DOPA-coated substrate maintains a strong adhesion strength even after 50 cycles of successive adhesion test (above 4 kPa, Supplementary Fig. 9b).

**Working mechanism for reversible wet adhesion**. To elucidate the basic mechanism responsible for the reversible wet adhesion, we next characterize the adhesion strength in response to temperature trigger in dry condition. At both room temperature and 40 °C, there is no marked difference in the adhesion strength (Supplementary Fig. 10), which is in striking contrast to the reversible adhesion observed in wet environment. In wet environment at room temperature, the pNIPAM side chains on

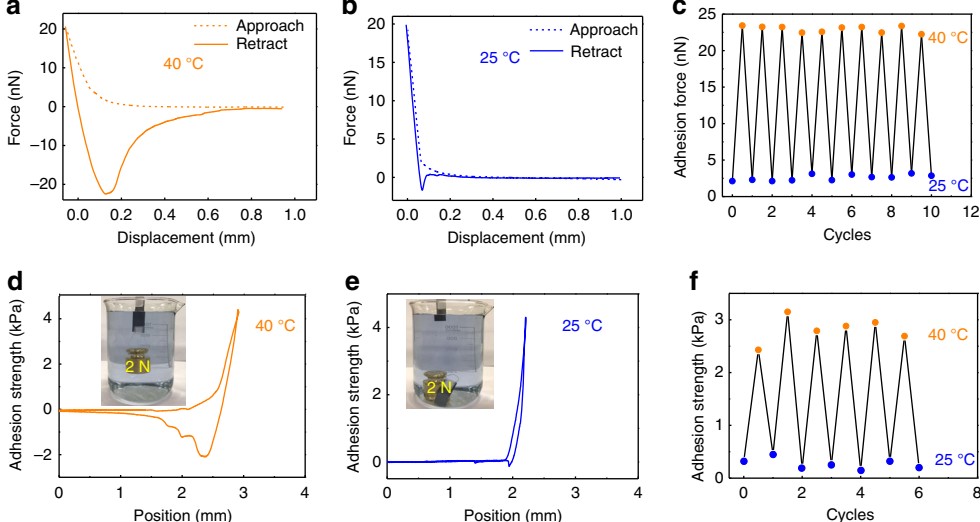

**Fig. 2** Characterization of wet adhesion. **a**, **b** Atomic force microscope (AFM) measurement of the wet adhesion force of the as-fabricated adhesive at 40 °C (**a**) and 25 °C (**b**), respectively. **c** The demonstration of reversible control of underwater adhesion by tuning the surface temperature. **d**, **e** Macroscopic measurement of the wet adhesion strength between the as-fabricated adhesive and silicon substrate at 40 °C (**d**) and 25 °C (**e**), respectively. **f** The demonstration of reversible underwater adhesion by controlling the local temperature of the adhesive

adhesive surface can easily form intermolecular hydrogen bonding with adjacent water molecules[40, 41] (Fig. 4a). As a result, the infusion of water molecules transforms the pNIPAM into a swelling state. Moreover, as a result of the host–guest chemistry, the adhesive moiety DOPA is spatially confined underneath the swelling pNIPAM side chains, thereby, the interfacial interaction between DOPA and target surface is dramatically screened. By contrast, when exposed to an external temperature above LCST, the intramolecular hydrogen bonding of the pNIPAM with adjacent water molecules is broken down (Fig. 4b). The collapse of pNIPAM side chains is further evidenced by the change in the thickness of the adhesive coating (Supplementary Fig. 11). In the wet environment, the thickness of the adhesive coating at 25 °C is 115 nm, which is much larger than that at 40 °C (90 nm). The collapse of the host copolymer is also associated with the marked variation in the global wettability (Methods section and Supplementary Fig. 12). As shown in Supplementary Fig. 13, at 40 °C, the static water CA on the adhesive coating increases to ~82°, which is in striking contrast to the hydrophilic property (~41°) at room temperature. Accordingly, the adhesive moiety DOPA is reexposed, therefore, the interfacial interaction is activated again. Thus, the wet adhesion activity can be dynamically regulated by reversibly screening/activating the interfacial interaction using a simple temperature trigger.

## Discussion
To demonstrate the versatility of our wet adhesive, we also quantify the adhesion strength of the as-synthesized coating against wide-ranging solid substrates including inorganic (glass, silicon, titanium, and aluminum) and organic surfaces (PDMS, PTFE). As shown in Fig. 5a, the wet adhesion strength on various substrates is comparable, suggesting that the interfacial adhesion is independent of target materials. Moreover, the wet adhesion strength can be further amplified by depositing the as-prepared adhesive onto gecko-like surfaces (Methods section and Supplementary Fig. 13). Figure 5b shows the SEM image of patterned PDMS post arrays with post diameter and height of 5 and 10 μm, respectively. Unlike the gecko which loses its adhesive property in water or high-humidity surrounding[42, 43], our experimental results show that the adhesive strength on compliant post arrays

at both 25 and 40 °C is much larger than that on the flat surface owing to the enlarged effective contact area.

Finally, we demonstrate the utility of the as-synthesized adhesive for the controlled pick and place of an object in an aqueous environment. As shown in Fig. 5c, by adjusting the local adhesive temperature above or below LCST, the interfacial adhesion can be activated or screened, allowing a metal block of ~200 g to be transported (Supplementary Movie 1) and released at any preferential location (Supplementary Movie 2). The setting time to switch the adhesion for pick and place can be arbitrarily controlled to fit specific applications. Moreover, as discussed earlier, the LCST of our adhesive can be predicted and modulated without a marked alternation of the adhesive property. As shown in Fig. 5d, the LCST can be changed between 32 and 40 °C by controlling different pNIPAM to CD monomer proportion. Thus, the tunability in the switching temperature, adhesion strength, as well as the setting time to rectify the adhesion can be leveraged to engineer bioadhesives that are capable of automatically responding to environmental stimuli, thus yielding smarter synthetic materials that satisfy application-specific requirements. Moreover, such an adhesive can be deposited to curved and flexible substrates and small flying microrobots for on-demand perching and adhesion without the cost and up-scaling concerns to achieve appealing functions which are unrealized using the conventional glues[19–21].

## Methods
**Chemicals**. Dopamine hydrochloride (DOPA-HCl, 98%), methacrylate anhydride (MAA, 95%), methoxyethyl acrylate (MEA, 98%), 1-adamantanecarboxylic acid chloride (98%), 2,2′-azobisisobutyronitrile (AIBN, 98%), hexane (99%), tetrahydrofuran (THF), and anhydrous dimethyl formamide (DMF) are purchased from Sigma-Aldrich. Sodium borate, ethyl acetate, sodium bicarbonate, magnesium sulfate, and sodium hydroxide are purchased from Meyer. Before utilization, methacrylate anhydride is passed through a column packed with $Al_2O_3$ to remove the inhibitor, and AIBN is recrystallized twice from methanol.

**Synthesis of DOPA monomer**. The DOPA monomer was synthesized from dopamine-HCl according to a previously described procedure with slight modifications[10]. Briefly, 10 g of sodium borate and 4 g of $NaHCO_3$ were first dissolved in 100 ml of deionized water and bubbled with $N_2$ for 20 min. Then, 5 g of dopamine-HCl (26.4 mmol) was added, followed by the dropwise addition of 4.7 ml of MEA (29 mmol) in 25 ml of THF, during which the pH of solution was kept above 8 with the addition of 1 M NaOH. The reaction mixture was stirred

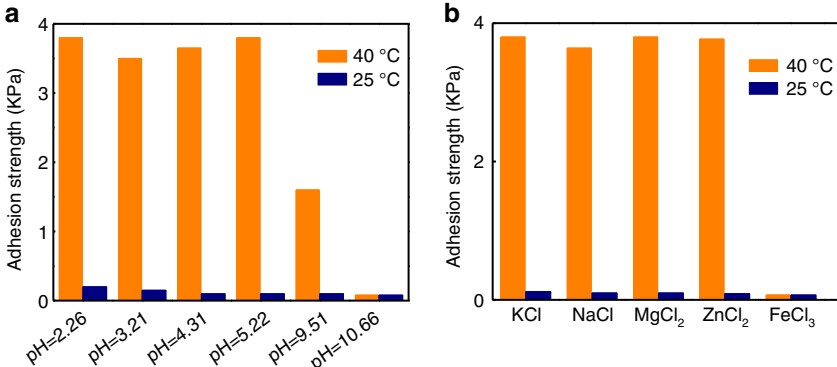

**Fig. 3** pH and ion effects on the wet adhesion. **a** The adhesion strength of the adhesive coating in solutions with smaller pH values such as 2.26, 3.21, 4.32, and 5.22 is comparable. Notably, there is a 50% reduction in the adhesion strength when the pH value of the solution is increased to 9.51. Moreover, for the coating immersed in the solution with pH of 10.66, there is no pronounced adhesion between two surfaces. **b** At 40 °C, the adhesion strength of the adhesive coating in 1 M NaCl, 1 M KCl, 1 M MgCl₂, and 1 M ZnCl₂ is determined to be ~4 kPa. However, for the adhesive immersed in the solution of 1 M FeCl₃, there is no measurable adhesion

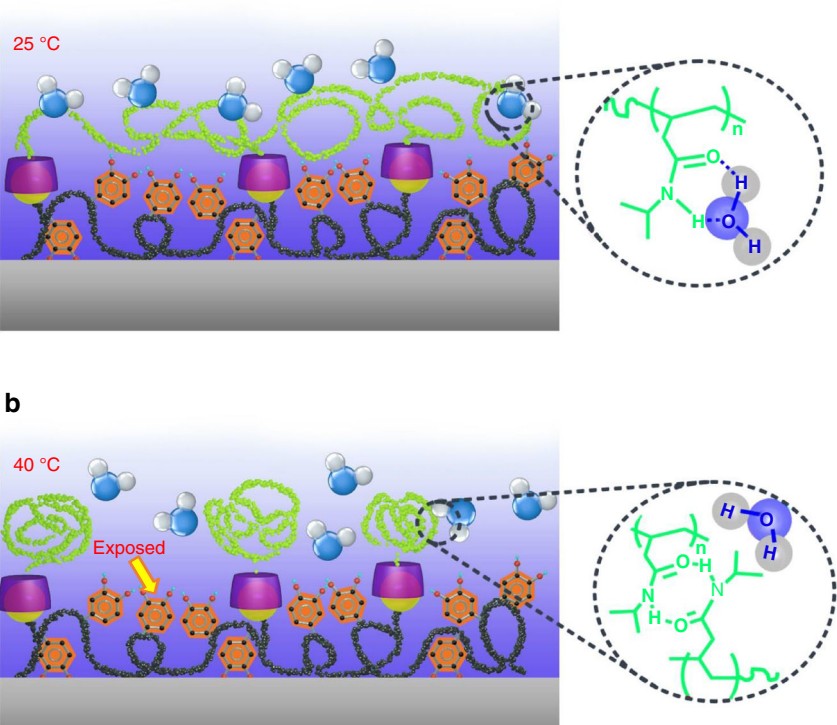

**Fig. 4** Working mechanism. **a** Schematic drawing shows the screening of the interfacial adhesion when the local temperature of the adhesive is below LCST. In this case, the pNIPAM can easily form intermolecular hydrogen bonding with adjacent water molecules and the infused water layer transforms the pNIPAM side chains to a swelling layer. As a result of selective recognition between the host and guest copolymers, the adhesive moiety DOPA is spatially stabilized and confined underneath the swelling pNIPAM chains, screening the interfacial interaction of the adhesive moiety DOPA with target surface. **b** The collapse of pNIPAM-CD chains is also associated with the formation of numerous agglomerates during the phase transition. As a result, the adhesive group is exposed, leading to pronounced interfacial adhesion

overnight at room temperature with N₂ bubbling. After being washed twice with 50 ml of ethyl acetate and adjusting the pH of less than 2, the aqueous solution was extracted with 50 ml of ethyl acetate three times. The as-obtained ethyl acetate layers were then combined and dried over MgSO₄ to a volume of 25 ml. Then, 200 ml of hexane was added with vigorous stirring and the suspension was held at 4 °C overnight to obtain the crude product. Finally, after drying and purification by recrystallization from hexane, we obtained the as-synthesized DOPA monomer. The yield of the DOPA monomer is 67%.

**Synthesis of AD monomer**. The AD monomer was prepared with adamantane-carboxylic acid chloride in anhydrous methylene chloride. Specifically, 30 ml of anhydrous methylene chloride was first added to a 200-ml round-bottom flask as a solvent and bubbled with N₂ at 0 °C for 20 min. Subsequently, 5.94 g of ada-mantanecarboxylic acid chloride and 5 ml of triethylamine were dissolved separately in the degassed anhydrous methylene chloride. Then, 5.216 g of 2-hydroxyethyl methacrylate (0.04 mol) was dissolved into 20 ml of anhydrous methylene chloride, and then added into the reaction solution dropwise. Subsequently, the reaction mixture was agitated for 4 h at 0 °C. The crude material was

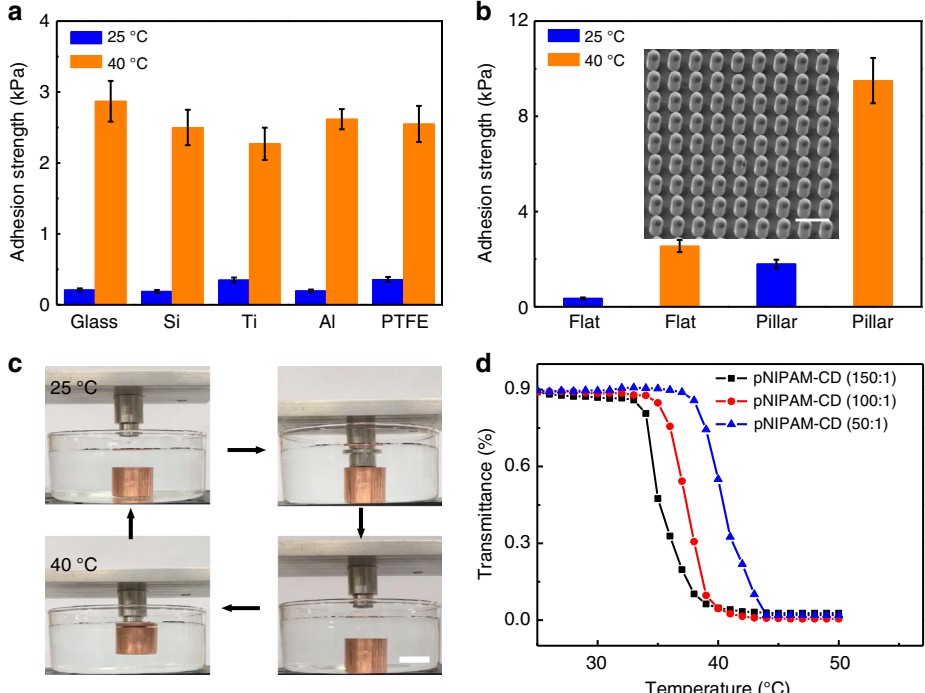

**Fig. 5** Versatility of reversible wet adhesive. **a** The adhesion strength of thermal-sensitive underwater adhesive copolymer system on various substrates under different temperatures indicating that the adhesion is applicable. **b** The combination of the wet adhesive with rough PDMS pillar arrays to amplify the adhesion strength. The scale bar is 15 μm. **c** The demonstration of the utility of as-fabricated adhesive for the well-controlled pick and place of a solid object (200 g) in the water environment. Scale bar, 2 cm. **d** The variation of LCST under different pNIPAM to CD monomer proportion in the host copolymer (pNIPAM-CD)

dissolved in hydrochloric acid (100 ml), followed by washing with saturated sodium carbonate solution until reaching a neutral pH. After removing the methylene chloride using a rotary evaporator, we purified the as-obtained product by Biotage Isolera TM Prime automatic column chromatography (Biotage SNAP 50 g of silica column; methylene chloride/methyl alcohol 10:1 to a gradient; flow rate 1440 ml min⁻¹). The yield of the as-synthesized AD monomer is calculated to be 85%.

**Synthesis of pDOPA-AD-MEA.** The adhesive guest copolymer pDOPA-AD-MEA was synthesized via free radical copolymerization of DOPA monomer, AD monomer, and MEA in a nitrogen environment (Supplementary Fig. 1a). In detail, 1 g of DOPA monomer, 4 g of MEA, and 1.5 g of AD monomer were dissolved separately in 15 ml of degassed DMF in a 100-ml round-bottom flask under the atmosphere of nitrogen to protect the product from oxidation. The solution was stirred at room temperature for 20 min, followed by the addition of 60 mg of AIBN. After degassing for three times (freeze–vacuum–thaw–nitrogen purging), the reaction mixture was agitated for 12 h at 80 °C and then DMF was removed using a rotary evaporator. The crude product was dissolved in 5 ml of THF, and purified by three repeats of precipitation in 50 ml of ethyl alcohol at 4 °C using a centrifuge at 8000 rpm for 30 min. Thus, we obtained the pDOPA-AD-MEA copolymer. The molecular weight of the as-synthesized pDOPA-AD-MEA copolymer is 8522 g mol⁻¹ and the yield is 63%.

**Synthesis of pNIPAM-CD.** Host copolymer pNIPAM-CD was prepared by the free radical polymerization of N-isopropylacrylamide and amino-β-cyclodextrin with AIBN as the initiator (Supplementary Fig. 1b). Briefly, 2.26 g of N-isopropylacrylamide and 30 mg of glycidyl methacrylate were dissolved separately in 10 ml of degassed DMF in a 50-ml Schlenk flask under nitrogen protection and stirred at room temperature for 1 h to obtain a homogeneous solution. Subsequently, 20 mg of AIBN was added to the solution under the protection of nitrogen. After 4 h of polymerization at 80 °C, 0.2 g of amino-β-cyclodextrin was further added into the reaction solution and agitated for 10 h at 80 °C for the copolymerization of guest copolymer pNIPAM-CD. After purification in 30 ml of hot water using a centrifuge at 10,000 r.p.m. for 20 min, we obtained the final product of pNIPAM-CD copolymer. The molecular weight of the as-synthesized pNIPAM-CD copolymer is 47,535 g mol⁻¹ and the yield is 85%.

**Preparation of adhesive coating.** We chose silicon as the substrate for the deposition of adhesive coating. After a thorough cleaning in Piranha solution containing sulfuric acid (97% $H_2SO_4$) and hydrogen peroxide, the silicon substrate was dip coated in ethanol solution of pDOPA-AD-MEA (5 mg ml⁻¹) for ~20 min at 70 °C. After immersion of the silicon surface into pNIPAM-CD solution (5 mg ml⁻¹) 30–120 min, a host copolymer pNIPAM-CD was achieved.

**Characterizations.** The copolymer composition was determined by 1H NMR (400 MHz) analysis using Varian VNMRS 400-MHz spectrometer in dimethyl sulfoxide (DMSO). The molecular weights of the copolymers were determined by a WATERS 1515 Series gel permeation chromatography (GPC) analyzer (concentration 5 mg ml⁻¹, THF eluent). Surface element component was analyzed by X-ray photoelectron spectroscopy (XPS, ESCALAB 250Xi multifunctional spectrometer, Thermo Fisher) using Al Kα radiation. Attenuated total reflection infrared (ATR-IR) spectrum was measured using a Nicolet iS10 instrument (Thermal Nicolet Corporation). UV–vis absorption spectrum of the pNIPAM-CD is recorded on UV2600 spectrometer (SHIMADZU).The sessile water droplet contact angle (CA) measurement is conducted using a DSA-100 optical contact angle meter (Kruss Company, Ltd., Germany) at 25 and 40 °C, respectively. Scanning electron microscope (SEM) images were obtained on a JSM-6701F field emission scanning electron microscope (FE-SEM) at 5–10 kV. The microscopic underwater adhesion property was quantified using an AFM integrated with a temperature controller (Bruker Optics). The macroscopic adhesion was measured using a high-sensitive universal testing machine (UTM, SHIMADZU, EZ-LX).

**Adhesion tests.** We measured the microscopic wet adhesion using an AFM integrated with a temperature controller, which allows for the simultaneous measurement of the adhesive under different temperature conditions. In a typical adhesion experiment, the AFM tip (Bruker, NP-S10 $Si_3N_4$ tip) was brought into contact with the adhesive coating, and then retracted. The force applied to separate the tip from the adhesive coating surface was determined as the adhesion force. The adhesion strength was determined using universal testing machine (UTM) (EZ-LX, Shimadzu Scientific Instruments.) equipped with a 200 N load cell (Supplementary Fig. 5). Specifically, the crosshead speed was set to be 10 mm min⁻¹ in the tensile mode to obtain the load-displacement curve. To provide a wet environment, we deigned a small chamber to encapsulate the adhesive and target surfaces. Briefly, a silicon substrate decorated with the adhesive coating was tightly fixed to the as-designed chamber and the other thoroughly cleaned silicon substrate was used as the target surface. The applying load was set as 1 N and the contact time between

the catechol surface and the target surface was 0 s. After being compressed for 0 s with a load of 1 N, two surfaces were separated and the interfacial adhesion strength can be determined based on the adhesion force curve. The local temperature of the adhesive coating was controlled by the temperature controller, which was fixed on the measurement equipment. As a comparison, we also measured the underwater adhesion strength of the commercially available 3 M double-sided tape (Supplementary Fig. 6). The work of adhesion of our adhesive coating in wet environment at 40 °C is calculated by using the method reported in previous studies[33, 34]. Briefly, the work of adhesion ($W_{adh}$) is quantified as the ratio of the integration of the adhesion force ($F$) vs. displacement ($z$) curves relative to the contact area, expressed as:

$$W_{adh} = \frac{\int F \, dz}{s} \qquad (1)$$

By contrast, the work of adhesion of our adhesive coating in underwater environment at 25 °C is 182 mJ m$^{-2}$, which is one order of magnitude smaller than that at 40 °C (4370 mJ m$^{-2}$).

**Preparation of the PDMS micropillar arrays**. The PDMS pillar arrays are fabricated using typical soft photolithography process (Supplementary Fig. 10). Briefly, we first fabricated a silicon template with microhole structure using soft photolithography and deep reactive ion etching (DRIE) processes. After functionalizing the as-fabricated silicon template using trichloro (1H, 1H, 2H, 2H-perfluoro-octyl) silane vapor for 1 h, we then poured PDMS on it, followed by curing at 100 °C for 2 h. Finally, we obtained PDMS pillar arrays by lift-off process. The diameter, center-to-center spacing, and height of the PDMS pillars are 5, 10, and 10 μm, respectively.

**Data availability**. The data that support the findings of this study are available on request from the corresponding authors (Z.W. or F.Z.).

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

## Acknowledgements

We are grateful for financial support from Research Grants Council of Hong Kong (no. 11275216 and no.11213915), National Natural Science Foundation of China (no. 51475401, no. 21434009, and no. 51605470), City University of Hong Kong (no. 9667139 and no. 9360140), Shenzhen Scientific and Innovation Bureau (no. JCYJ2017041314141208098), National Key Research and Development Program of China (2016YFC1100401), and Chinese Academy of Sciences (no. QYZDY-SSW-JSC013).

## Author contributions

Z.W. and F.Z. supervised the research. Y.Z., Y.W., and L.W. designed the experiment. Y.Z. and Y.W. prepared the samples. Y.W., Y.Z., and L.W. conducted the microscale and macroscale characterization. Z.W., Y.Z., Y.W., F.Z., and J.L. wrote the paper and all the authors discussed the manuscript.

## Additional information

**Competing interests:** The authors declare no competing financial interests.

