## [Peer Review File · Nature Communications]

Reviewers' comments:

Reviewer #1 (Remarks to the Author):

This manuscript reported a novel strategy to fabricate an underwater smart adhesive coating, whose adhesive strength could be easily activated and deactivated repeatedly by tuning the environmental temperature. The proposed mechanism is that a thermo-responsive polymer was coupled to the adhesive coating via host-guest interaction. The coupled polymer could either screen or activate the adhesive effect by its temperature-induced hydrophilicity change. This work is interesting and has great potential in various practical applications. However, there are quite a bit of missing information.

Questions:

Is this specific host-guest interaction between CD and AD reversible or permanent? If it's reversible, what condition would cause the dissociation? What is the strength of this interaction compared to that of the catechol-surface interaction?

Is this wet-resistant adhesive strength expected to have pH or ionic strength dependence?

Both methoxyethyl acrylate and methacrylate anhydride were abbreviated to MEA, please distinguish them.

Need more information in the protocol. For example, the authors do not specify the yield of the monomers or polymers that were synthesized. NMR and FTIR is not sufficient in the characterization of the polymer. What are the molecular weights of these polymers?

Adhesion testing methods. The author should include information such as the rate of approach and detachment during adhesion testing. The author should calculate work of adhesion.

The author used dopamine instead of DOPA throughout the paper. There is a clear difference between these two different compounds.

For the array of micropillars, how was it made and what were the distances between pillars?

What are the thicknesses of the coating?

Reviewer #2 (Remarks to the Author):

The manuscript entitled "Bio-inspired reversible underwater adhesive" by Yanhua Zhao et al. describes the temperature-triggered reversible adhesive system that works in wet environment. The concept of reversible wet adhesive is interesting and the ability to "pick-and-place" objects by using only the chemistry and difference in temperature is intriguing. The adhesive is simple and straightforward in design but the manuscript shows only a superficial evidence on how the adhesion works (Figure 3 and related text), which makes it hard to confirm whether this adhesive behavior is based on what they claimed or not. In addition, the target applications are unclear and also the adhesion force is quite low, so it seems to require perspectives on how this adhesive technology can be useful in certain applications under water. Please see below for the suggested changes for publication.

1. The adhesion mechanism is not clear. The pNIPAAm is on the outermost surface and there is no background or explanation on why the polymer is considered adhesive in high temperature, and

how they can be reversible in adhesion. Without the mechanism or rationale of the polymer design, the manuscript lacks proper evidence on the claims.

2. In addition to the comment 1, if the catechol groups are responsible for the adhesion as described in page 5 and 6, they should also have adhesion in hydrated state because dopa has strong adhesion in water. Having said that, it is not clear how they lose adhesion and release an object as mentioned in Figure 3c and Video 2. It requires an in-depth study on the release mechanism too.

3. The manuscript lacks any particular applications or at least suggestions of the fields that this might be useful for.

4. At the end of the manuscript, authors mentioned "such an adhesive can be deposited to curved and flexible substrates and small flying micro-robots for on-demand perching...". For small flying micro-robots, it seems require adhesion in dry state. Have you checked dry adhesion as well? And how the dry adhesion would work using DOPA?

5. In Figure 4c, weight of the object is not mentioned anywhere.

6. The TEM images in Figure 3 should be in the same scale.

7. The detailed procedures for TEM image acquisition is not described. Did you use any staining to visualize exclusively pNIPAM? The TEM images look typical artifact from drying process and it is very odd that you could exclusively saw pNIPAM although you have a layer of pDOPA-AD-MEA as well. Having said that TEM doesn't look a very good method to demonstrate the transition from "homogenous" to "agglomerates". Also, there are agglomerates in Figure 3b as well and it seems the matter of contrast difference. Spatial chemical mapping should be helpful to show the evidence for this (e.g. TOF-SIMS etc.).

Reviewer #3 (Remarks to the Author):

This article describes a bicomponent adhesive system that works underwater. The polymers contains DOPA groups that allow deposition of the first polymer film on different substrates, host-guest groups that allow deposition of the second polymer by reversible interactions, and PNIPAM as T sensitive polymer able to collapse or swell with T change and therefore forming a soft hydrogel or a rigid insoluble polymer film with adhesive differences. A possible contribution of embedded DOPA groups to adhesion underwater is suggested, though not proven.

The principle is interesting, though there are many claims in this manuscript that require more careful experiments for scientific approval. The performance of the system at macroscopic scale is rather disappointing. The limits of the system related to stability of DOPA groups, pH and ionic concentration ranges where activity is maintained etc, have to be properly discussed.

Although the concept is of acceptable novelty and broad interest, the manuscript does not seem to be mature. In the following text I raise major points to be considered in a corrected version.

Polymer synthesis: the molecular weight of the polymers should be characterized. Catechols are radical scavengers, so low molecular weight of the DOPA containing polymer is expected. Is this the case? Is this relevant for the application? How much catechol was introduced in the polymer chains? Does the catechol fraction in the copolymer correspond to the monomer ratio in the polymerization mixture? The same question should be answered for the other monomers.

Adhesion testing: conditions for testing are poorly described in the SI. Pulling speed, contact time

etc.... These parameters play an important role in the final value of adhesion force

Page 2: "the attainment of the reversible adhesion in the wet environment remains extremely elusive due to the complex interfacial interaction" Authors should include other ways to revert adhesion in catechol containing adhesives, like for example the use of photoactivatable nitrodopamine (Shafiq et al, Angew Chem 2012)

Catechols are rather instable and, assuming that they play a role in this system, their stability will be crucial for maintaining the reversibility of the system. The stability of the catechols is dependent on pH and oxidant. Authors should comment on this as possible limitation of their system.

Authors use a hydrogel as adhesive ("high" adhesion is measured when PNIPAM is in swollen form), so the adhesive will have low cohesiveness and this will limit adhesion strength. Authors should comment on this

Figure 1d, caption: "The AT-IR spectrum of the adhesive coating at 1630 cm⁻¹, 1525 cm⁻¹, 1380 cm⁻¹ and 1365 cm⁻¹ are assigned to double methyl groups on pNIPAM, which demonstrates the presence of DOPA and pNIPAM, respectively". Also in the manuscript text "Manuscript: In addition, the ATIR spectra clearly show the presence of the characteristic peaks of DOPA and pNIPAM (Fig. 1d), which further demonstrates the successful synthesis of the adhesive coating." Authors did not assign any of the signals to the catechol groups, so how does this demonstrate their presence? The catechol groups are not contained in the PNIPAM polymer. The properties of the adhesive film are poorly characterized. Thickness of each film in dry and wet conditions at the two temperatures, and topography (roughness) underwater (AFM) at the two temperatures is mandatory. TEM pictures are taken in dry state, so it does not help much.

"Taken together, compared to previous approach using chemical control which entails a long response time, the adhesion in our adhesive can be dynamically mediated in a more flexible and faster manner." Which previous approach? If authors refer to previously reported work, reference should be added

Characterization in dry conditions. Authors state that there is no adhesion difference at room temperature and at 40°C. This is not surprising. A dry hydrogel will not be adherent. Moreover, drying of the catechol containing film might well lead to reactions between catechol groups and might lead to irreversible changes in the adhesive properties. Authors should check and report on this. The sentence "This suggests that the wet adhesion should be intricately hinging upon the interaction with the water phase" does not really have a scientific meaning and should be deleted.

The discussion about adhesion mechanism at molecular level (Paragraph in pages 5-6) is fully unclear and has quite some scientific inconsistencies and missinterpretations:

- "...the infused water serves as a lubricating film and transforms the pNIPAM into a swelling state". Lubrication is different from swelling, and is not necessarily a relevant mechanism in this case. Water swells the layer and, consequently, the different groups present in the hydrogel film are able to interact with the countersurface.
- "...a swelling state with a global hydrophilic property with a water CA of 41°C" This measurement does not really add any information at this point. Obviously a hydrogel is hydrophilic. I assume authors want to describe the differences of the hydrogel film in swollen and collapse state with T. Authors should rephrase so that this becomes more clear.
- "Indeed, as shown in Fig.3b, TEM measurement also reveals that the host pNIPAM-CD displays a homogeneous state without the occurrence of aggregation". The meaning of this sentence at this point is unclear. I guess authors want to say that the swollen film forms a homogeneous, flat film, as they show in the TEM pictures (though TEM pictures are taken in dry state) vs the rougher surface observed in the collapsed state at 40°C. What is meant with "indeed"?

- Figure 3 should include the temperatures to which the represented mechanism and TEM correspond. The word "screening" is misleading and inappropriate.
- „Moreover, as a result of the specific host-guest chemistry, the adhesive moiety DOPA is spatially confined and stabilized underneath the swelling pNIPAM side-chains, thereby the interfacial interaction between DOPA and target surface is dramatically screened" This is an assumption, there is no proof for that. Authors should either provide a proof or reformulate this statement
- "Accordingly, the adhesive moiety DOPA re-emerges from the pNIPAM, re-activating the interfacial interaction." This is again an educated guess, but there is no proof of that. In fact, this proof is absolutely necessary for this paper. Authors should either perform experiments with the film without DOPA or with DOPA groups protected or blocked so that they are not able to interact with countersurface.
- "The intramolecular transformation at the supramolecular level is also associated with the marked variation in the wettability at a global scale (see Methods)." What is meant with "intramolecular transformation at supramolecular level"? I guess collapse of the PNIPAM containing polymer as a consequence of surpassing LCST? Then this should be named "collapse".
- "Thus, the wet adhesion activity can be dynamically regulated by screening or activating the interfacial interaction using a simple temperature trigger" What is meant with "screening". This wording is incorrect and misleading

In summary of all these comments: the working mechanism is not properly investigated. It is fully unclear if the DOPA groups contribute to adhesion or only serve to anchor the polymer to the surface. Moreover, catechol groups are highly reactive, so it is unclear for how long they remain available for interactions. Experiments with blocked catechol groups are necessary to confirm their contribution to the adhesion strength.

"The wet adhesion strength on various substrates are comparable, suggesting that the interfacial adhesion is not independent of target materials." This part is fully unclear. I assume the authors have coated different surfaces using their polymer. This does not mean that their adhesive would work with any countersurface, which is what is claimed in their sentence. In fact, this would be highly unexpected considering the type of interactions expected for this system and described above. Or did the authors measured adhesive strength against PTFE, glass, etc as countersurfaces? Authors seem to mix different concepts. Universality is a big claim for an adhesive, so authors need to justify properly.

The videos demonstrating the macroscopic performance of the adhesive film are rather disappointing. Where is the CLAIMED reversibility shown? What about pick and place?

Caption Video 2: "supramolecular shielding effect" WHAT IS THAT? Authors should be more careful with fancy expressions and call things by their proper names.

Point by Point Response to Referees' Comments

Reviewer #1 (Remarks to the Author):

This manuscript reported a novel strategy to fabricate an underwater smart adhesive coating, whose adhesive strength could be easily activated and deactivated repeatedly by tuning the environmental temperature. The proposed mechanism is that a thermo-responsive polymer was coupled to the adhesive coating via host-guest interaction. The coupled polymer could either screen or activate the adhesive effect by its temperature-induced hydrophilicity change. This work is interesting and has great potential in various practical applications. However, there are quite a bit of missing information.

Response: We thank the referee very much for his/her high comment on the novelty of our work.

1. Is this specific host-guest interaction between CD and AD reversible or permanent? If it's reversible, what condition would cause the dissociation? What is the strength of this interaction compared to that of the catechol-surface interaction?

Response: We thank the referee very much for this insightful question. The specific host-guest interaction between CD and AD is non-covalent and is based on supramolecular chemistry. As one of the most investigated host and guest pairs, CD and AD host-guest interaction can form strong and permanent host-guest binding with an association constant around 1500 M^{-1} in water, which is almost the strongest binding between CD and its guest molecules (1,2). Although the host-guest interaction in our adhesive system is not reversible, it indirectly contributes to the reversible wet adhesion by tuning external temperature. Briefly, the use of host-guest interaction enables a robust linking between adhesive polymer (pDOPA-AD-MEA) and thermal-sensitive polymer (pNIPAM-CD). Thus, when the local temperature is above lower critical solution temperature (LCST), the adhesive moiety is exposed and gives rise to a superior adhesion (We will elaborate this point in Fig. R4).

2. Is this wet-resistant adhesive strength expected to have pH or ionic strength dependence?

Response: This is a very interesting and important point. In practical applications, the underwater adhesive might be operated under different conditions. Therefore, we investigated how the adhesion strength is influenced by pH and ion types of solutions. We first conducted the adhesion measurement in solutions with pH values ranging from 2.26 to 10.66. During the experiment, the adhesive coating was first deposited on a silicon substrate (1.5 cm*1.5 cm) and then compressed with a silicon target substrate with an applying load of 1 N in solutions. Figure R1 a and b show typical adhesion force curves under these solutions at 40 °C and 25 °C, respectively. Based on the adhesion force curve, the adhesion strength can be determined (Figure R1c). As shown in Figure R1c, the adhesion properties of the adhesive coating in solutions with smaller pH values such as 2.26, 3.21, 4.32, and 5.22 are comparable. Notably, there is a 50% reduction in the adhesion strength when the pH value of the solution increases to 9.51. Moreover, for the coating immersed in the solution with pH of 10.66, there is no pronounced adhesion between two surfaces. These results clearly suggest that the adhesion property of our adhesive is intricately sensitive to pH. The failure of wet adhesion in alkaline solutions might be due to the oxidation of the adhesive catechol according to references (3, 4). At

room temperature, the interfacial adhesion is deactivated and thus it is independent on the pH of solutions (Figure R1 b and c).

Figure R1 | pH effect on the wet adhesion. The adhesion force curve of the adhesive coating under various pH solutions at 40 °C (a) and 25 °C (b), respectively. (c) The plot of the adhesion strength at 25 °C and 40 °C in various pH conditions.

To investigate whether or how the ions mediates the adhesion strength of the adhesive coating, we further characterized the adhesion behaviors of our adhesive in various ionic solutions including 1M NaCl, 1M KCl, 1M MgCl₂, 1M ZnCl₂, and 1M FeCl₃. Based on the adhesion force curves shown in Figure R2a and b, we plot the variation of the adhesion strength at different solutions and different temperatures (Figure R2c). At 40 °C, the adhesion strength of the adhesive coating in 1M NaCl, 1M KCl, 1M MgCl₂ and 1M ZnCl₂ is about 4 kPa, which is approximately identical with that in pure water. However, for the adhesive immersed in the solution of 1M FeCl₃, there is no measurable adhesion. Such a loss of wet adhesion strength at solution with Fe³⁺ might be ascribed to the selective binding of Fe³⁺ with DOPA moiety and subsequent formation of Fe(DOPA)₃ cross-links, which has been verified in references (5, 6, 7). Taken together, we demonstrate that our adhesive coating can maintain a superior wet adhesion in a wide range of pH and ion windows, though it becomes ineffective at solutions with Fe³⁺ ion or high pH. Related discussion has been included in the revised manuscript (*Page 5, Line 19 - Page 6, Line 9*).

Figure R2 | Ion effect on the wet adhesion. The adhesion force curve of the adhesive coating under various ionic solutions at 40 °C (a) and 25 °C (b), respectively. (c) The variation of adhesion strength in different ionic solutions.

3. Both methoxyethyl acrylate and methacrylate anhydride were abbreviated to MEA, please distinguish them.

Response: We thank the referee for pointing out this omission. In our revised manuscript, we have referred to methoxyethyl acrylate and methacrylate anhydride as MEA and MAA, respectively. Please see *Page 1, line 17-18* in the revised supporting information.

4. Need more information in the protocol. For example, the authors do not specify the of the monomers or polymers that were synthesized. NMR and FTIR is not sufficient in the characterization of the polymer. What are the molecular weights of these polymers?

Response: The yields of DOPA monomer, AD monomer, pDOPA-AD-MEA copolymer, and pNIPAM-CD are 67%, 85%, 63%, and 85%, respectively (see *Page 2, Line 6, Line 19, Page3, Line5, Line16* in the supporting information). Based on the Gel Permeation Chromatography (GPC) measurement, the molecular weights of the guest copolymer pDOPA-AD-MEA and host copolymer pNIPAM-CD are determined to be 8522 and 47535 g/mol, respectively (see *Page 3, Line 4-5, and Line 15-16* in the supporting information).

5. Adhesion testing methods. The author should include information such as the rate of approach and detachment during adhesion testing. The author should calculate work of adhesion.

Response: The adhesion testing was conducted using Universal Testing Machine (UTM, EZ-LX, Shimadzu Scientific Instruments) equipped with a 200 N load cell. The rate of approach and detachment was 10 mm·min⁻¹ in tensile mode. The applying load was set as 1 N and the contact time between the adhesive coating and target surface was 0 s (see *Page 5, Line 9-23* in the supporting information). The adhesion strength was calculated as the ratio of the applied load relative to the contact area (*s*). The work of adhesion of our adhesive coating in wet environment at 40 °C is calculated to be 4370 mJ/m² by using the method reported in previous studies (8, 9). Briefly, the work of adhesion (W_{adh}) is quantified as the ratio of the integration of the adhesion force (F) versus displacement (z) curves relative to the contact area, expressed as:

$$W_{adh} = \frac{\int Fdz}{s}$$

By contrast, the work of adhesion of our adhesive coating in underwater environment at 25 °C is 182 mJ/m², which is one order magnitude smaller than that at 40 °C. Detailed descriptions on the testing method and work of adhesion have been included in the revised manuscript (*Page 5, Line 4-6*).

6. The author used dopamine instead of DOPA throughout the paper. There is a clear difference between these two different compounds.

Response: We agree with the referee that dopamine and DOPA are different chemical compounds, which correspond to 3,4-dihydroxyphenethylamine and 3,4-dihydroxy-L-phenylalanine, respectively. In our work, we use DOPA as the adhesive moiety to synthesize the adhesive guest copolymer pDOPA-AD-MEA.

7. For the array of micropillars, how was it made and what were the distances between pillars?

Response: The PDMS pillar arrays are fabricated using typical soft photolithography process. Briefly, we first fabricated a silicon template with micro-hole structure using soft photolithography and deep reactive ion etching (DRIE) processes (Figure R3). After functionalizing the as-fabricated silicon template using trichloro(1H,1H,2H,2H-perfluoro-octyl) silane vapor for 1 h to make the surface hydrophobic, we then poured PDMS on it, followed by curing at 100 °C for 2 h. Finally, we obtained PDMS pillar arrays by lift-off process. The diameter, center-to-center spacing, height of the PDMS pillars are 5, 10, 10 μm, respectively. We have included the detailed fabrication process into the supporting information in the revised manuscript (see *Page6, Line 17-23* in the supporting information).

Figure R3 | Schematic diagram showing the detailed process to fabricate PDMS pillar arrays. A silicon template with micro-hole arrays was first fabricated using photolithography and deep reactive ion etching. Then PDMS was cast onto the as-fabricated template, followed by curing and lift-off.

8. What are the thicknesses of the coating?

Response: The thickness of the coating can be tailored easily. To facilitate the adhesion measurement using AFM, the thickness of the adhesive coating was kept very thin (~120 nm). For the macroscopic adhesion test, we prepared all the adhesive coatings with the thickness of ~ 90 μm using the scraping method.

Reviewer #2 (Remarks to the Author):

The manuscript entitled “Bio-inspired reversible underwater adhesive” by Yanhua Zhao et al. describes the temperature-triggered reversible adhesive system that works in wet environment. The concept of reversible wet adhesive is interesting and the ability to “pick-and-place” objects by using only the chemistry and difference in temperature is intriguing. The adhesive is simple and straightforward in design but the manuscript shows only a superficial evidence on how the adhesion works (Figure 3 and related text), which makes it hard to confirm whether this adhesive behavior is based on what they claimed or not. In addition, the target applications are unclear and also the adhesion force is quite low, so it seems to require perspectives on how this adhesive technology can be useful in certain applications under water. Please see below for the suggested changes for publication.

Response: We are pleased that the referee finds our work interesting and intriguing. We are also grateful to have this opportunity to clarify and substantiate the working mechanisms of our novel adhesive.

1. The adhesion mechanism is not clear. The pNIPAAm is on the outermost surface and there is no background or explanation on why the polymer is considered adhesive in high temperature, and how they can be reversible in adhesion. Without the mechanism or rationale of the polymer design, the manuscript lacks proper evidence on the claims.

Response: Let's first discuss the rationale of the design of our adhesive materials. As shown in Figure R4, the underwater adhesive consists of two copolymers: the guest adhesive copolymer AD-MEA-DOPA and the host thermo-responsive copolymer pNIPAM-CD. For the guest adhesive copolymer, AD is the guest moiety which binds with the host moiety (CD), DOPA is the adhesive moiety that is expected to provide wet adhesion at a wide range of temperature, and MEA is the hydrophobic moiety which serves as a matrix for binding with the adhesive moiety. In the host copolymer, CD is the host moiety to link guest polymer, and pNIPAM is the thermo-responsive moiety. We choose pNIPAM because it can display reversible conformational transition from expanded wire to compact globule in response to an external temperature trigger in wet environment. Note that the wet adhesion results from the DOPA adhesive moiety instead of pNIPAM.

In our experiment, the copolymer AD-MEA-DOPA is designed to deposit on solid substrate (Figure R4a), followed by the assembly of thermo-responsive copolymer pNIPAM-CD via the host-guest recognition between CD moiety and AD moiety (Figure R4b). At room temperature, pNIPAM-CD can absorb water molecules in wet environment and become into a swelling layer, which in turn spatially covers the adhesive moiety (Figure R4c). The inflated pNIPAM layer serves as a screening barrier that prevents the direct contact between the adhesive moiety and the inherent surface, hence deactivate the adhesion property of our adhesive coating. By contrast, when exposed to external temperature above LCST, the pNIPAM-CD exhibits a conformational transition and collapses into compact globule. Accordingly, the adhesive DOPA moiety is exposed and the adhesion property is activated again (Figure R4d).

Figure R4 | Schematic diagram showing the mechanism of the thermo-responsive underwater adhesion. (a, b) The schematic diagram showing the procedure for the preparation of the wet adhesive. The synthesis process involves the creation of the adhesive guest copolymer pDOPA-AD-MEA on a clean Si substrate using “dip coating” method (a), followed by the self-assembly of host copolymer pNIPAM-CD using the host-guest molecular recognition (b). (c) Schematic drawing shows the screening of the interfacial adhesion when the local temperature of the adhesive is below LCST. In this case, the pNIPAM can easily form intermolecular hydrogen bonding with adjacent water molecules and the infused water layer transforms the pNIPAM side-chains to a swelling state.

As a result of selective recognition between the host and guest copolymers, the adhesive moiety DOPA is spatially confined underneath the swelling pNIPAM chains, screening the interfacial interaction of the adhesive moiety DOPA with target surface. **(d)** When the local temperature is above LCST, the pNIPAM side-chains collapse and the adhesive moiety DOPA is re-exposed, leading to pronounced interfacial adhesion again.

The above mentioned mechanisms are convincingly proved by our experiments. First, we compared the adhesion properties between the wet adhesive coating and control sample. In our control experiment, we prepared a guest copolymer AD-MEA without the incorporation of DOPA, followed by the assembly of host copolymer pNIPAM-CD via the function of host-guest recognition. As illustrated in Figure R5a, we find that the adhesion strength of the control sample is ~ 1.8 kPa, which is much smaller than that of the wet adhesive coating (AD-MEA-DOPA). Moreover, for the control sample, a significant decay in the adhesion strength can be observed with successive testing cycles. After ~ 50 cycles of test, the adhesion strength almost decreases to zero (Figure R5a). However, our wet adhesive maintains a strong adhesion strength after 50 cycles of successive tests (above 4 kPa, Figure R5b). Our result is consistent with previous study by Haeshin Lee (10).

Second, the reversibility in the adhesion is ascribed to the conformational transition of pNIPAM-CD triggered by the temperature stimulus, which can screen or activate the adhesive moiety on demand. Figure R6 shows the AFM image of the adhesive coating surface at 25 °C and 40 °C, respectively. Careful comparison of the AFM images at different temperatures reveals that the homogenous state of the adhesive surface at 25 °C is transformed into a heterogeneous surface covered by globule (Figure R6). The conformational transition is further evidenced by the measurement of the thickness of the adhesive coating using AFM (Figure R7). In the wet environment the thickness of the adhesive coating at 25 °C is ~ 115 nm, which is larger than that at 40 °C (~ 90 nm). By contrast, in the dry state, there is no marked difference in the thickness of the adhesive coating in response to different temperatures (Figure R7 b). This is because in such dry state, the host copolymer pNIPAM-CD stays in a coiled condition.

Figure R5| Comparison of wet adhesion strength between AD-MEA-coated substrate and AD-MEA-DOPA coated substrate at 40°C. (a) For the control sample without the use of DOPA (AD-MEA), the adhesion strength exhibits a significant decay during successive tests. The curves in black, red, blue, pink, green and navy blue correspond to the test conducted on 1st, 11th, 21st, 31st, 41st, and 51st cycle, respectively. **(b)** By contrast, for the adhesive made of AD-MEA-DOPA, there is no marked decay in the adhesion strength even after 50 cycles of tests.

Figure R6| Morphology measurement of adhesive coating using AFM at 25 °C (a) and 40 °C (b), respectively. (a) The AFM image shows the homogenous dispersion of the pNIPAM-CD without the occurrence of aggregation at 25 °C. **(b)** When the temperature is above LCST, intramolecular hydrogen bonding inside the pNIPAM chains is formed, reactivating the appearance of compact globule.

Figure R7| Thickness measurement of the adhesive coating in wet (a) and dry (b) states. (a) In the wet environment, the thickness of the adhesive coating at 25 °C is ~115 nm, which is larger than that at 40 °C (~90 nm). (b) In the dry environment, the thickness of the adhesive coating at 25 °C is 80 nm, which is almost the same as that at 40 °C.

2. In addition to the comment 1, if the catechol groups are responsible for the adhesion as described in page 5 and 6, they should also have adhesion in hydrated state because dopa has strong adhesion in water. Having said that, it is not clear how they lose adhesion and release an object as mentioned in Figure 3c and Video 2. It requires an in-depth study on the release mechanism too.

Response: Catechol groups are responsible for the adhesion in a wet environment. In hydrated state where the pNIPAM exhibits a swelling property, the catechol groups are spatially covered by the swelling pNIPAM. In such a state, the adhesion becomes negligible. Please also see our response to comment 1 as discussed above.

3. The manuscript lacks any particular applications or at least suggestions of the fields that this might be useful for.

Response: The focus of our manuscript is to report a new adhesive which can reversibly switch its adhesion in a wet environment in response to local temperature change. In our paper, we have shown the pick-place application. Such a facile approach to dynamically switch the adhesion might find applications such as surgery operation or wound dressings. Inspired by recent exciting studies which report the development of octopus-inspired wet resistant adhesive or slug-inspired tough adhesive (11, 12), we believe that our adhesive will find lots of applications, which will be reported in our future studies.

4. At the end of the manuscript, authors mentioned “such an adhesive can be deposited to curved and flexible substrates and small flying micro-robots for on-demand perching...” For small flying micro-robots, it seems require adhesion in dry state. Have you checked dry adhesion as well? And how the dry adhesion would work using DOPA?

Response: We agree with the referee that the state of the art flying micro-robots mainly operate in dry state. With the rapid development in robotics and bio-inspired engineering, we envision that robots

which can freely move in the water will emerge soon. In addition to the wet adhesion, our adhesive also exhibits superior dry adhesion. The adhesion strength of the coating in dry state is measured to be ~20 kPa (Figure R8). But in this case, the adhesion is not reversible in response to the temperature change.

Figure R8| Adhesion strength of the adhesive coating in the air. The adhesion strength of the adhesive coating at 25 °C and 40 °C in air is ~20 kPa, demonstrating that the thermo-reversible adhesion cannot be achieved in dry condition.

5. In Figure 4c, weight of the object is not mentioned anywhere.
Response: The weight of the object in Figure 4c is ~ 200 g and we have added the weight information of the object in the revised manuscript (*Page 13, Line 7*).

6. The TEM images in Figure 3 should be in the same scale.
Response: We have replaced the TEM images with AFM images in the revised supporting information (*Page 13, Supplementary Figure. 11*).

7. The detailed procedures for TEM image acquisition is not described. Did you use any staining to visualize exclusively pNIPAM? The TEM images look typical artifact from drying process and it is very odd that you could exclusively saw pNIPAM although you have a layer of pDOPA-AD-MEA as well. Having said that TEM doesn't look a very good method to demonstrate the transition from "homogenous" to "agglomerates". Also, there are agglomerates in Figure 3b as well and it seems the matter of contrast difference. Spatial chemical mapping should be helpful to show the evidence for this (e.g. TOF-SIMS etc.).

Response: The TEM image was taken with the cryo-dried method. In detail, a copper grid coated with carbon was dipped into the pNIPAM-CD solution and put into a vial, which was plunged into liquid nitrogen. Then, water was removed from the frozen specimen by a freeze-drier and the original state of pNIPAM-CD can be maintained. The TEM image is exclusively taken from pNIPAM-CD, which does not contain the layer of pDOPA-AD-MEA.

We thank the referee very much for proposing different spatial chemical mapping methods. Although we could not conduct TOF-SIMS measurement due to the lack of this special equipment in our institutions, our AFM measurement does provide an alternative approach to convincingly confirm the conformational transition of the adhesive coating from "homogenous" to "agglomerates" when the temperature changes from room temperature to 40 °C (Figure R 6), as we discussed earlier.

Reviewer #3 (Remarks to the Author):

This article describes a bicomponent adhesive system that works underwater. The polymers contains DOPA groups that allow deposition of the first polymer film on different substrates, host-guest groups that allow deposition of the second polymer by reversible interactions, and PNIPAM as T sensitive polymer able to collapse or swell with T change and therefore forming a soft hydrogel or a rigid insoluble polymer film with adhesive differences. A possible contribution of embedded DOPA groups to adhesion underwater is suggested, though not proven. The principle is interesting, though there are many claims in this manuscript that require more careful experiments for scientific approval. The performance of the system at macroscopic scale is rather disappointing. The limits of the system related to stability of DOPA groups, pH and ionic concentration ranges where activity is maintained etc, have to be properly discussed.

Response: We thank the referee very much for pointing out that our work is novel and of broad interest. The main contribution and focus of our work is to develop a new adhesive which can dynamically, reversibly switch its coating in wet environment, which has never been reported before. By saying that, we argue that our adhesive also exhibits excellent adhesion at macroscopic level.

The relatively small macroscopic adhesion is mainly owing to the small applying load and contact time we applied. In our experiment, the applying load onto two substrates was set as 1N and the contact time was set as 0 s, which are smaller in comparison with other previous studies (12). To substantiate our claim, we measured the adhesive force in wet conditions under different applying load and contact time (Figure R9a and c). Figure R9 b and d plot the variation of adhesion behavior as a function of pre-applying load and contact time. We find that the adhesion strength and adhesion force increased with the augment of the applying load and contact time. For an applying load 1 N and contact time of 120 s, the adhesion strength increased to 18 kPa, which is comparable to those reported in other studies (12).

Figure R9| Tailoring underwater adhesion by pre-applying load and contact time. (a, b) The variation of the adhesion strength under different applying loads at 40 °C. (c, d) The variation of the adhesion strength under different contact time at 40 °C.

1. Polymer synthesis: the molecular weight of the polymers should be characterized. Catechols are radical scavengers, so low molecular weight of the DOPA containing polymer is expected. Is this the case? Is this relevant for the application? How much catechol was introduced in the polymer chains? Does the catechol fraction in the copolymer correspond to the monomer ratio in the polymerization mixture? The same question should be answered for the other monomers.

Response: Following the reviewer's suggestion, we have characterized the molecular weight of the guest copolymer pDOPA-AD-MEA and host copolymer pNIPAM-CD by Gel Permeation Chromatography (GPC), which are 8522 g/mol and 47535 g/mol, respectively (Please see *page 3, line 4-5* and *line 15-16* in the revised supporting information). To our best knowledge, catechol itself is not radical scavenger. However, when there is oxygen in the reaction environment, catechol can be oxidized to quinone which is a kind of radical scavenger. During our experiment, the copolymerization reaction is conducted under nitrogen environment, which prevents catechol from turning into scavenger. Thus, we believe that catechol does not much affect the polymerization as radical scavenger.

In the polymerization mixture, the monomer ratio of AD, MEA, and DOPA was set as 14.9%, 74.5%, and 10.6% and the monomer ratio of NIPAM and CD was 95.8% and 4.12% respectively. Based on the ^1H NMR measurements, we determined the contents of two copolymers. For guest polymer AD-MEA-DOPA, the monomer ratio of AD, MEA, and DOPA in the copolymer was 29.6%, 51.2%, and 18.5%, respectively. The decline in the monomer ratio of MEA fraction in the copolymer is ascribed to the evaporation during the reaction process. For the host copolymer pNIPAM-CD, the monomer ratio of NIPAM and CD was 95.6% and 4.3%, respectively, which is almost the same with the monomer ratio during the reaction.

2. Adhesion testing: conditions for testing are poorly described in the SI. Pulling speed, contact time etc.... These parameters play an important role in the final value of adhesion force
Response: The adhesion strengths were determined using Universal Testing Machine (UTM) (EZ-

LX, Shimadzu Scientific Instruments), equipped with a 200 N load cell. The crosshead speed was set to $10 \text{ mm}\cdot\text{min}^{-1}$ in tensile mode to obtain the load-displacement curve. The applying load was set as 1 N and the contact time between the catechol surface and the target surface was 0 s. The testing conditions for the testing have been added in the supporting information of the revised manuscript (Please see **Pages 5, line 9-23** in the supporting information).

3. Page 2: “the attainment of the reversible adhesion in the wet environment remains extremely elusive due to the complex interfacial interaction” Authors should include other ways to revert adhesion in catechol containing adhesives, like for example the use of photoactivatable nitrodopamine (Shafiq et al, Angew Chem 2012)

Response: We thank the reviewer very much for pointing out this important work, which has been cited in our revised manuscript (*ref 27*).

4. Catechols are rather instable and, assuming that they play a role in this system, their stability will be crucial for maintaining the reversibility of the system. The stability of the catechols is dependent on pH and oxidant. Authors should comment on this as possible limitation of their system.

Response: We thank the reviewer for this insightful comment. Indeed, we find that the underwater adhesion is sensitive to pH value of solutions. We have done a series of experiments to characterize the pH or ion dependence of the adhesive coating (Figure R 10 and 11). We find that the adhesive film is quite stable in solutions with pH lower than 9. Yet, oxidants like hydrogen peroxide would destroy the adhesion property (7). Related discussion on the instability of the adhesive due to the oxidant has been included in our revised manuscript (**Page 5, Line 19 - Page 6, Line 9**). The main focus of this paper is to report a facile approach for the design of novel underwater adhesive. In the future, further optimization will be conducted to make this adhesive closer to practical applications.

Figure R10 | pH effect on the wet adhesion. The adhesion force curve of the adhesive coating under various pH solutions at 40 °C (a) and 25 °C (b), respectively. (c) The plot of the adhesion strength at 25 °C and 40 °C in various pH conditions.

Figure R11| Ion effect on wet adhesion. The adhesion force curve of the adhesive coating under various ionic solutions at 40 °C (a) and 25 °C (b), respectively. (c) The variation of adhesion strength in different ionic solutions.

5. Authors use a hydrogel as adhesive (“high” adhesion is measured when PNIPAM is in swollen form), so the adhesive will have low cohesiveness and this will limit adhesion strength. Authors should comment on this.

Response: We agree that an ideal adhesive demands both high cohesiveness and good interfacial interaction. In our design, we have used the relatively hydrophobic (less hydrated) pDOPA-AD-MEA as guest copolymer, and pNIPAM-CD as the host polymer to tune the exposure of DOPA moiety. At high adhesion state, pNIPAM-CD would shrink to non-hydrated polymer. Thus, both guest and host polymer would appear as less hydrated states, therefore the adhesive coating itself won’t break down. As discussed in the manuscript, our adhesive exhibits superior reversible adhesion even after many cycles of measurement. Still, in the future, we will continue to optimize our design to make our adhesive suitable for practical applications.

6. Figure 1d, caption: “The AT-IR spectrum of the adhesive coating at 1630 cm⁻¹, 1525 cm⁻¹, 1380 cm⁻¹ and 1365 cm⁻¹ are assigned to benzene group and double methyl groups on pNIPAM, which demonstrates the presence of DOPA and pNIPAM, respectively”. Also in the manuscript text “Manuscript: In addition, the ATIR spectra clearly show the presence of the characteristic peaks of DOPA and pNIPAM (Fig. 1d), which further demonstrates the successful synthesis of the adhesive coating.” Authors did not assign any of the signals to the catechol groups, so how does this demonstrate their presence? The catechol groups are not contained in the PNIPAM polymer.

Response: We thank the reviewer for pointing out the mistake, and we have corrected the description in the revised manuscript as follows:

Page 9, Line 12–Page 10, Line 1-3: “The AT-IR spectrum of the adhesive coating at 1630 cm⁻¹, 1525 cm⁻¹, 1380 cm⁻¹ and 1365 cm⁻¹ are assigned to the C-H of aromatic ring on DOPA and double methyl groups on pNIPAM, which demonstrates the presence of DOPA and pNIPAM, respectively”.

7. The properties of the adhesive film are poorly characterized. Thickness of each film in dry and wet conditions at the two temperatures, and topography (roughness) underwater (AFM) at the two temperatures is mandatory. TEM pictures are taken in dry state, so it does not help much.

Response: We thank the reviewer very much for the important suggestion on presenting more results on the characterization of our adhesive coating, which has been supplemented in the revised manuscript. Briefly, we measured the thickness of the adhesive film in both dry and wet conditions, and characterized the topography (roughness) underwater with AFM (Figure R12). We find that the underwater roughness of the adhesive film at 40 °C is 0.586 nm which is smaller than that at 25 °C (1.06 nm). Meanwhile, the thickness of the adhesive coating at 25 °C is larger than that at 40 °C in the wet environment (Figure R13a). Obviously, the thermo-sensitive conformational transition of pNIPAM-CD lies behind such phenomena. In detail, at 25 °C, pNIPAM-CD can absorb water and exhibit a swollen state filled with water which increased both the roughness and thickness of the adhesive film. However, when temperature increased to 40 °C, the pNIPAM-CD expelled water and transformed into the state of compact globule, which is responsible for the smaller surface roughness and the decrease of the surface thickness. In the dry state with the absence of water, the host copolymer pNIPAM-CD stays in a coiled conformation, so no difference on the film thickness can be observed when changing temperature (Figure R13b).

Figure R12| Morphology measurement of adhesive coating using AFM at 25 °C (a) and 40 °C (b), respectively. (a) At room temperature, AFM measurement shows a homogenous dispersion of the pNIPAM-CD in the surface, without the occurrence of aggregation. **(b)** When the temperature is above LCST, the surface is heterogeneous covered by globule.

Figure R13| Thickness measurement of the adhesive coating in wet and dry state. (a) In wet environment, the thickness of the adhesive coating at 25 °C is 115 nm, which is larger than that at 40 °C (90 nm). (b) In dry environment, the thickness of the adhesive coating at 25 °C is 80 nm, which is almost the same as that at 40 °C.

8. “Taken together, compared to previous approach using chemical control which entails a long response time, the adhesion in our adhesive can be dynamically mediated in a more flexible and faster manner.” Which previous approach? If authors refer to previously reported work, reference should be added.

Response: We thank the reviewer for pointing out the omission. We have cited related references in the revised manuscript (*Ref 4* and *Ref 14*).

9. Characterization in dry conditions. Authors state that there is no adhesion difference at room temperature and at 40°C. This is not surprising. A dry hydrogel will not be adherent. Moreover, drying of the catechol containing film might well lead to reactions between catechol groups and might lead to irreversible changes in the adhesive properties. Authors should check and report on this. The sentence “This suggests that the wet adhesion should be intricately hinging upon the interaction with the water phase” does not really have a scientific meaning and should be deleted.

Response: Our adhesive is a sticky polymer at room temperature, and it is not hydrogel. Our adhesive coating consists of two parts, the relatively hydrophobic (less hydrated) pDOPA-AD-MEA serves as the adhesive guest copolymer, and pNIPAM-CD as the host polymer to tune the exposure of DOPA moiety. The guest copolymer was first applied to a solid substrate followed by the self-assembly of the host copolymer. In wet environment, at room temperature, the upper film pNIPAM-CD is hydrophilic and it will absorb water and exhibit as a swollen layer on the top of the adhesive guest copolymer pDOPA-AD-MEA. In this case, the swelling pNIPAM-CD layer serves as a barrier which prevents the direct contact between the adhesive guest copolymer and the target surface, hence the adhesion is screened. However, at a temperature above LCST, water molecules are expelled out and pNIPAM-CD exhibits in a coiled state. Thus, the adhesive copolymer is re-exposed and the wet adhesion is activated again. Note that in dry state at both room temperature and 40°C, the adhesive exhibit a strong adhesion strength of ~ 20 kPa, and no reversibility of the adhesion can be observed.

To avoid potential misunderstanding, we have deleted the sentence “This suggests that the wet adhesion should be intricately hinging upon the interaction with the water phase” in the revised manuscript. It should be noted that the catechol containing adhesive is a less hydrated film, and the drying did not lead to irreversible changes in the adhesive properties.

10. The discussion about adhesion mechanism at molecular level (Paragraph in pages 5-6) is fully unclear and has quite some scientific inconsistencies and miss-interpretations:

Response: We thank the reviewer for pointing out our omission in the discussion of the adhesion mechanism at molecular level, and we would like to discuss the thermo-responsive underwater adhesion in a more detailed manner.

Let’s first discuss the rationale underlying the design of our adhesive materials. As shown in Figure R14, the underwater adhesive consists of two copolymers: the guest adhesive copolymer AD-MEA-DOPA and the host thermo-responsive copolymer pNIPAM-CD. For the guest adhesive copolymer, AD is the guest moiety which binds with the host moiety (CD), DOPA is the adhesive moiety that is expected to provide wet adhesion at a wide range of temperature, and MEA is the hydrophobic moiety which serves as a matrix for binding with the adhesive moiety. In the host copolymer, CD is the host moiety to link guest polymer, and pNIPAM is the thermo-responsive moiety. We choose pNIPAM because it can display reversible conformational transition from expanded wire to compact globule in response to an external temperature trigger (around its lower critical solution temperature (LCST)) in the wet environment. Note that the wet adhesion is attributed to the DOPA adhesive moiety instead of pNIPAM.

In our experiment, the copolymer AD-MEA-DOPA is designed to deposit on any solid substrates (Figure R14a), followed by the assembly of thermo-responsive copolymer pNIPAM-CD via the host-guest recognition between CD moiety and AD moiety (Figure R14 b). At room temperature, pNIPAM-CD can absorb water molecules in wet environment and become into a swelling layer, which in turn spatially cover the adhesive moiety (Figure R14 c). The inflated pNIPAM layer serves as a screening barrier that prevents the direct contact between the adhesive moiety and the inherent surface, hence deactivate the adhesion property of our adhesive coating. By contrast, when exposed to external temperature above LCST, the pNIPAM-CD exhibits a conformational transition and collapses into compact globule. Accordingly, the adhesive DOPA moiety is re-exposed and the adhesion property is activated again (Figure R14 d).

Figure R14 | Schematic diagram showing the mechanism of the thermo-responsive underwater adhesion. (a, b) The schematic diagram showing the procedure for the preparation of the wet adhesive. The synthesis process involves the creation of the adhesive guest copolymer pDOPA-AD-MEA on a clean Si substrate using “dip coating” method (a), followed by the self-assembly of host copolymer pNIPAM-CD using the host-guest molecular recognition (b). (c) Schematic drawing shows

the screening of the interfacial adhesion when the local temperature of the adhesive is below LCST. In this case, the pNIPAM can easily form intermolecular hydrogen bonding with adjacent water molecules and the infused water layer transforms the pNIPAM side-chains to a swelling state. As a result of selective recognition between the host and guest copolymers, the adhesive moiety DOPA is spatially confined underneath the swelling pNIPAM chains, screening the interfacial interaction of the adhesive moiety DOPA with target surface. **(d)** When the local temperature is above LCST, the pNIPAM side-chains collapse and the adhesive groups are re-exposed, leading to pronounced interfacial adhesion again.

The above mentioned mechanisms are convincingly proved by our experiments. First, we compared the adhesion properties between the wet adhesive coating and control sample. In our control experiment, we prepared a guest copolymer AD-MEA without the incorporation of DOPA, followed by the assembly of host copolymer pNIPAM-CD via the function of host-guest recognition. As illustrated in Figure R 15a, we find that the adhesion strength of the control sample (AD-MEA) is ~ 1.8 kPa, which is much smaller than that of the wet adhesive (AD-MEA-DOPA). Moreover, a significant decay in the adhesion strength can be observed with successive testing cycles. After ~ 50 cycles of test, the adhesion strength almost decreases to zero (Figure R15a). However, our wet adhesive maintains a strong adhesion strength after 50 cycles of successive tests (above 4 kPa, Figure R15 b). Our result is consistent with previous study by Haeshin Lee (10).

Figure R15| Wet adhesion comparison between AD-MEA-coated substrate and AD-MEA-DOPA coated substrate at 40°C. **(a)** A significant decay in adhesion was observed with successive contacts for the AD-MEA coated substrate. The adhesion strength shown in the figure represent every 10th cycle: 1(1.8 kPa, black), 11(1.6 kPa, red), 21(1 kPa, blue), 31(0.1 kPa, pink), 41(0.1 kPa, green), and 51(0.1 kPa, navy blue). The rapid decay of wet adhesion strength may be caused by the loss of AD-MEA polymer. Small amount of AD-MAE was taken away when the upper pair detached from AD-MAE coating. **(b)** The adhesion strength of AD-MEA-DOPA coated substrate is strong with even successive contacts, after 50 cycles of adhesion test experiment, the adhesion strength of AD-MEA-DOPA coated substrate is still above 4 kPa.

Second, the reversibility in the adhesion is ascribed to the conformational transition of pNIPAM-CD triggered by the temperature stimulus, which can screen or activate the adhesive moiety on

demand. Figure R16 shows the AFM image of the adhesive coating surface at 25 °C and 40 °C, respectively. Careful comparison of the AFM images at different temperatures reveals that the homogenous state of the adhesive surface at 25 °C is transformed into a heterogeneous surface covered by globule (Figure R16). The conformational transition is further evidenced by the measurement of the thickness of the adhesive coating using AFM (Figure R17). In the wet environment the thickness of the adhesive coating at 25 °C is 115 nm, which is larger than that at 40 °C (90 nm). By contrast, in the dry state, there is no marked difference in the thickness of the adhesive coating in response to different temperatures (Figure R17 b). This is because in such dry state, the host copolymer pNIPAM-CD stays in a coiled condition.

Such information has been discussed in the revised manuscript (*Page 6, Line 10-25,-Page 7, Line 1-19*).

Figure R16| Morphology measurement of adhesive coating using AFM at 25 °C (a) and 40 °C (b), respectively. (a) The AFM image shows the homogenous dispersion of the pNIPAM-CD without the occurrence of aggregation at 25 °C. (b) For a temperature above LCST, intramolecular hydrogen bonding inside the pNIPAM chains is formed, reactivating the appearance of compact globule.

Figure R17| Thickness measurement of the adhesive coating in wet state. (a) In wet environment, the thickness of the adhesive coating at 25 °C is ~115 nm, which is larger than that at 40 °C (90 nm).

(b) In the dry environment, the thickness of the adhesive coating at 25 °C is ~80 nm, which is almost the same as that at 40 °C.

11. "...the infused water serves as a lubricating film and transforms the pNIPAM into a swelling state". Lubrication is different from swelling, and is not necessarily a relevant mechanism in this case. Water swells the layer and, consequently, the different groups present in the hydrogel film are able to interact with the counter surface.

Response: At room temperature, pNIPAM can transform into a swelling layer filled with water which can serve as a lubricating layer and a natural barrier that protect adhesive moiety from the contact with target surface. The definition of lubrication is utilized here to emphasize the isolating effect of the water filled swelling layer, which is inappropriate. Such description has been deleted in the revised manuscript.

12 "...a swelling state with a global hydrophilic property with a water CA of 41°C" This measurement does not really add any information at this point. Obviously a hydrogel is hydrophilic. I assume authors want to describe the differences of the hydrogel film in swollen and collapse state with T. Authors should rephrase so that this becomes more clear.

Response: In our previous submission, we used the contact angle measurement to indirectly quantify the morphology transformation of the host copolymer in response to the temperature trigger. In wet environment and at room temperature, the pNIPAM-CD is hydrophilic, and it will exhibit as a swollen blanket on top of the adhesive guest copolymer pDOPA-AD-MEA owing to the adsorption of water. In this case, the pNIPAM-CD blanket serves as a barrier, blocking the direct contact between the adhesive guest copolymer and the target surface. However, when the local temperature is raised above the LCST of pNIPAM-CD, pNIPAM-CD displays a hydrophobic property and pNIPAM-CD exists in a coiled state. As a result, the adhesive copolymer is exposed and the adhesion is reactivated. The change of the wettability in accordance with different temperature can indirectly testify our experiment mechanism (Please refer to *Page 6, Line 1-16* in the supporting information). However, we are open to the removal of these data if the referee feel they are not necessary.

13. "Indeed, as shown in Fig.3b, TEM measurement also reveals that the host pNIPAM-CD displays a homogeneous state without the occurrence of aggregation". The meaning of this sentence at this point is unclear. I guess authors want to say that the swollen film forms a homogeneous, flat film, as they show in the TEM pictures (though TEM pictures are taken in dry state not dry state) vs the rougher surface observed in the collapsed state at 40°C. What is meant with "indeed"?

Response: Following the reviewer's suggestion, we replaced the TEM pictures with AFM images to characterize the surface morphology of the adhesive coating in wet environment at different temperatures. Related discussions can be found in *Page 7, Line 7-14*.

14. Figure 3 should include the temperatures to which the represented mechanism and TEM correspond. The word "screening" is misleading and inappropriate.

Response: This comment is also related to the working mechanism of our adhesive, which has been elaborated in our response to Comment 10. The swelling of pNIPAM side-chains results in the screening of the interfacial interaction between two surfaces. Temperatures have been included in Figure 4 of the revised manuscript.

15. Moreover, as a result of the specific host-guest chemistry, the adhesive moiety DOPA is spatially confined and stabilized underneath the swelling pNIPAM side-chains, thereby the interfacial interaction between DOPA and target surface is dramatically screened” This is an assumption, there is no proof for that. Authors should either provide a proof or reformulate this statement

Response: In our design, the pNIPAM-CD is deposited on the top of pDOPA-AD-MEA copolymer. Although it is challenging to visualize the specific location of these copolymers, our experimental results do demonstrate that when the local temperature is above LCST, the interfacial adhesion between two surfaces is dramatically reduced. In the revised manuscript, we have deleted the word of “stabilized”.

16 “Accordingly, the adhesive moiety DOPA re-emerges from the pNIPAM, re-activating the interfacial interaction.” This is again an educated guess, but there is no proof of that. In fact, this proof is absolutely necessary for this paper. Authors should either perform experiments with the film without DOPA or with DOPA groups protected or blocked so that they are not able to interact with counter surface.

Response: This comment is also related to the working mechanism of our wet adhesive, which has been elaborated in our response to Comment 10. As discussed previously, for the control sample without the presence of DOPA, the adhesion strength largely dropped and the adhesion property would disappear as the test cycles increase (Figure R15a). By contrast, for the DOPA containing film the adhesion strength is ~ 4 kPa and it did not decrease as the testing cycles increase (Figure R15b). This experiment result indicates that the adhesion results from DOPA moiety in the adhesive coating.

17 A. “The intramolecular transformation at the supramolecular level is also associated with the marked variation in the wettability at a global scale (see Methods).” What is meant with “intramolecular transformation at supramolecular level”? I guess collapse of the PNIPAM containing polymer as a consequence of surpassing LCST? Then this should be named “collapse”.

Response: According to the reviewer’s suggestion, we have re-phrased the sentence as follows:

Page 7, Line 14-15 : “The collapse of the host copolymer is also associated with the marked variation in the global wettability”.

17B. “Thus, the wet adhesion activity can be dynamically regulated by screening or activating the interfacial interaction using a simple temperature trigger” What is meant with “screening”. This wording is incorrect and misleading

Response: This comment is also related to the working mechanism of our adhesive, which has been

discussed in our reply to Comment 10. Briefly, in hydrated state, the adhesive moiety is spatially covered by the swelling pNIPAM-CD and the adhesive moiety is spatially screened.

18. In summary of all these comments: the working mechanism is not properly investigated. It is fully unclear if the DOPA groups contribute to adhesion or only serve to anchor the polymer to the surface. Moreover, catechol groups are highly reactive, so it is unclear for how long they remain available for interactions. Experiments with blocked catechol groups are necessary to confirm their contribution to the adhesion strength.

Response: As discussed above, we have elaborated the detailed working mechanism responsible for the wet adhesion, which are fully supported by our experiments such as the characterization of the surface morphology and surface thickness using AFM and comparison of adhesion behaviors between the coatings with and without DOPA. New results have been included in the revised manuscript.

To evaluate the stability of the adhesive copolymer pDOPA-AD-MEA, we further characterized the time-dependent variation of the adhesion strength. During the experiment, the adhesive coating was immersed in wet environment with a temperature of 40°C. As shown in Figure R18, the adhesion strength of the adhesive coating is quite stable in the first 36 h. However, there is a marked drop in the adhesion strength after a continuous operation of 48 h. In the future, we will optimize our design to dramatically enhance the stability of our wet adhesive.

Figure R18| The plot of the time-dependent variation of the wet adhesion strength at 40°C.

19. “The wet adhesion strength on various substrates are comparable, suggesting that the interfacial adhesion is not independent of target materials.” This part is fully unclear. I assume the authors have coated different surfaces using their polymer. This does not mean that their adhesive would work with any counter surface, which is what is claimed in their sentence. In fact, this would be highly unexpected considering the type of interactions expected for this system and described above. Or did the authors measured adhesive strength against PTFE, glass, etc as counter surfaces? Authors seem to

mix different concepts. Universality is a big claim for an adhesive, so authors need to justify properly. **Response:** We thank the reviewer for pointing out the lack of clarity in our wording. We did conduct the adhesion measurement using the PTFE, glass, etc as counter surfaces. We found that the adhesion strength is comparable to that on silicon. We modified our description by deleting the wording of “universal” in the revised manuscript (*Page 7, Line 21*).

20. The videos demonstrating the macroscopic performance of the adhesive film are rather disappointing. Where is the CLAIMED reversibility shown? What about pick and place? **Response:** The videos demonstrate the adhesion performance of the adhesive coating at different temperatures. By adjusting the local adhesive temperature above or below LCST, the interfacial adhesion can be activated or screened, allowing a metal block ~200 g to be picked up and released. We have corrected the video’s name according to the reviewer’s suggestion (*Page 15, Line 12-15* in the supporting information).

21. Caption Video 2: “supramolecular shielding effect” WHAT IS THAT? Authors should be more careful with fancy expressions and call things by their proper names.

Response: In our previous submission, we used “shielding” as a substitute of screening. To make it clear, we have deleted the word of “shielding”.

References:

1. Harada, A., Kobayashi, R., Takashima, Y., Hashidzume, A., & Yamaguchi, H. Macroscopic self-assembly through molecular recognition. *Nat. Chem.* 31, 34-37 (2011).
2. Harada, A., Yoshinori, T., and Masaki, N., Supramolecular polymeric materials via cyclodextrin–guest interactions. *Acc. Chem. Res.* 477, 2128-2140 (2014).
3. Yu, J., Wei, W., Menyo, M. S., Masic, A., Waite, J. H., & Israelachvili, J. N. Adhesion of mussel foot protein-3 to TiO₂ surfaces: the effect of pH. *Biomacromolecules*.144, 1072-1077 (2013)
4. Cencer, M., Liu, Y., Winter, A., Murley, M., Meng, H., & Lee, B. P. Effect of pH on the rate of curing and bioadhesive properties of dopamine functionalized poly (ethylene glycol) hydrogels. *Biomacromolecules*. 158, 2861-2869 (2014).
5. Holten, A. N., et al. pH-induced metal-ligand cross-links inspired by mussel yield self-healing polymer networks with near-covalent elastic moduli. *Proc. Natl. Acad. Sci. U. S. A.* 108, 2651-2655 (2011).
6. Park, J. P., Song, I. T., Lee, J., Ryu, J. H., Lee, Y., & Lee, H. Vanadyl–Catecholamine Hydrogels Inspired by Ascidians and Mussels. *Chem. Mater.* 27, 105-111 (2014).
7. Monahan, J., Jonathan, J. W., Specificity of metal ion cross-linking in marine mussel adhesives. *Chem. Commun.* 14, 1672-1673 (2003).

8. Flanigan, C. M., Alfred, J. C., and Kenneth. R. S. Structural development and adhesion of acrylic ABA triblock copolymer gels. *Macromolecules*. 32, 7251-7262 (1999).
9. Narkar, A. R., Barker, B., Clisch, M., Jiang, J. & Lee, B. P. pH responsive and oxidation resistant wet adhesive based on reversible catechol–boronate complexation. *Chem. Mater.* 28, 5432-5439 (2016).
10. Lee, H., Lee, B. P. & Messersmith, P. B. A reversible wet/dry adhesive inspired by mussels and geckos. *Nature* 448, 338-341 (2007).
11. Li, J, et al. Tough adhesives for diverse wet surfaces. *Science*. 357, 378-381 (2017).
12. Baik, S, et al. A wet-tolerant adhesive patch inspired by protuberances in suction cups of octopi. *Nature*. 546, 396-400 (2017).

REVIEWERS' COMMENTS:

Reviewer #1 (Remarks to the Author):

The revision adequately addressed previous comments. I recommend it to be published.

Reviewer #3 (Remarks to the Author):

Authors have addressed all points raised. Manuscript is now in good shape for Nature Commun

Reviewer #4 (Remarks to the Author):

Authors have thoroughly revised the manuscript based on the previous comments. The new revisions strengthened the claims on adhesion mechanisms showing that the "guest" layer thickness changes in different temperatures by reversible hydration, and also showing that DOPA is responsible for the adhesion. All the changes are acceptable and there is one minor suggestion as below.

1. In the new AFM images, the height scale bar is different, so it is hard to make a fair comparison – (a) is 20nm and (b) is 5nm. In a close observation based on the colors of the AFM images, (a) seems to have more roughness and bigger agglomeration on the surface. The surface roughness does not necessarily have to change to make a thickness change, although the schematic might suggest that (i.e. in low temperature, it can be thick but still rough). The thickness change data in Figure S12 is already strong evidence to show it forms a thick layer on top of the DOPA layer. I suggest remove the AFM images.

Responses to the referees' comments

Reviewer #1 (Remarks to the Author):

The revision adequately addressed previous comments. I recommend it to be published.

Response: We thank the referee very much for the recommendation of the publication of our paper in Nature Communications.

Reviewer #3 (Remarks to the Author):

Authors have addressed all points raised. Manuscript is now in good shape for Nature Commun

Response: We thank the referee very much for the recommendation of the publication of our paper in Nature Communications.

Reviewer #4

Authors have thoroughly revised the manuscript based on the previous comments. The new revisions strengthened the claims on adhesion mechanisms showing that the “guest” layer thickness changes in different temperatures by reversible hydration, and also showing that DOPA is responsible for the adhesion. All the changes are acceptable and there is one minor suggestion as below.

1. In the new AFM images, the height scale bar is different, so it is hard to make a fair comparison – (a) is 20nm and (b) is 5nm. In a close observation based on the colors of the AFM images, (a) seems to have more roughness and bigger agglomeration on the surface. The surface roughness does not necessarily have to change to make a thickness change, although the schematic might suggest that (i.e. in low temperature, it can be thick but still rough). The thickness change data in Figure S12 is already strong evidence to show it forms a thick layer on top of the DOPA layer. I suggest remove the AFM images.

Response: We thank the referee very much for the recommendation of the publication of our paper. We fully agree with your suggestion regarding the removal of the AFM images from the manuscript, which has been done in the revised manuscript.